



# Development of a wireless, non-intrusive, MEMS-based pressure and acoustic measurement system for large-scale operating wind turbine blades

Sarah Barber[1], Julien Deparday[1], Yuriy Marykovskiy[1], Eleni Chatzi[2], Imad Abdallah[2], Gregory Duthé[2], Michele Magno[3], Tommaso Polonelli[3], Raphael Fischer[3], and Hanna Mueller[3]

[1]Institute of Energy Technology, Eastern Switzerland University of Applied Sciences, Oberseestrasse 10, 8640 Rapperswil, Switzerland
[2]Chair of Structural Mechanics and Monitoring, ETH Zürich, Switzerland
[3]D-ITET, Center for Project-Based Learning, ETH Zürich, Switzerland

**Correspondence:** Sarah Barber (sarah.barber@ost.ch); Imad Abdallah (abdallah@ibk.baug.ethz.ch)

**Abstract.** As the wind energy industry is maturing and wind turbines are growing, there is an increasing need for cost-effective monitoring and data analysis solutions to understand the complex aerodynamic and acoustic behaviour of the flexible blades. Published measurements on operating rotor blades in real conditions are very scarce, due to the complexity of the installation and use of measurement systems. However, recent developments in electronics, wireless communication and MEMS sensors are making it possible to acquire data in a cost-effective and energy-efficient way. In this work, therefore, a cost-effective MEMS-based aerodynamic and acoustic wireless measurement system that is thin, non-intrusive, easy to install, low power, and self-sustaining is designed and tested. The results show that the system is capable of delivering relevant results continuously, although work needs to be done on calibrating and correcting the pressure signals, as well as on refining the concept for the attachment sleeve for weather protection in the field. Finally, two methods for using the measurements to provide added value to the wind energy industry are developed and demonstrated: (1) inferring local angle of attack via stagnation point detection using differential pressure sensors near the leading edge, and (2) detecting and classifying leading edge erosion using instantaneous snapshots of the measured pressure fields. On-going work involves field tests on an operating 6 kW wind turbine in Switzerland.



# 1 Introduction

## 1.1 Full-scale aerodynamic measurements

As the wind energy industry is maturing and wind turbines are growing, there is an increasing need for cost-effective monitoring and data analysis solutions to understand the complex aerodynamic and acoustic behaviour of the flexible blades (Schepers and Schreck, 2019). The incoming flow transports turbulent structures of different scales spatially and temporally, yielding aerodynamic load fluctuations that are complex to simulate. A high shear flow due to the atmospheric boundary layer could create some additional instabilities. It can also change the relative wind speed and angle of attack at different heights of the rotor blades. These changes of the local inflow condition on the rotor blades contribute to non-linear aerodynamic loading. Even in steady conditions with well-known free-stream conditions, it is not easy to assess the local inflow conditions on a wind turbine, as the wind velocity decreases between the free-stream and the turbine rotor in a manner that varies with wind speed and rotational speed of the rotor. Adding to that, a span-wise component of the flow - creating a three-dimensional flow - makes it hard to correctly evaluate the local wind speed and angle of attack on the rotor blade, hence to compare it with simulations or measurements with a fixed blade.

Published aerodynamic and acoustic measurements on operating rotor blades in real conditions are very scarce, due to the complexity of installation and use of measurement systems. A well-known field experiment called DanAERO involved on-field experiments on a 2 MW wind turbine with an instrumented blade (Madsen et al., 2016; Troldborg et al.). In this project, the aerodynamic and acoustic properties of the wind turbine were thoroughly investigated, both in wind tunnel tests and in field tests. Far-field microphones were placed around the wind turbine and a blade was instrumented with 50 flush-mounted microphones to evaluate the noise emission of the blade and detect local flow separation. It was shown that such aero-acoustic field measurements have the potential to provide a high added value to the wind industry through furthered understanding of three-dimensional effects. However, the project required a very large effort and cost.

Before this, a set of wind turbine field experiments tackling the aerodynamics, the performance and noise emissions was carried out on an operating 2.3 MW wind turbine (Medina et al., 2011). In this work, a thorough characterisation of the inflow properties and of the structure of the wind turbine were investigated. For this, four five-hole pitot tubes were installed on the blade as well as 60-64 pressure taps at nine different locations along the span. An extended study was carried out to correct the pressure tap measurements in order to extract the most accurate local pressure measurements. Similarly to the DanAERO project, this study provided valuable information to the research community as well as to the industry, but a large amount of effort was required in order to instrument the wind turbine.

Even more recently, an on-field measurement system was developed for a 100 kW wind turbine (Wu et al., 2019). Similarly to previous experimental campaigns, pitot tubes were installed at different span-wise locations in order to evaluate the inflow conditions, and flush mounted pressure taps were installed at five different locations along the span and linked to pressure scanners thanks to tubes.

As well as demonstrating the potential value of aerodynamic and acoustic field measurements, all these measurement campaigns demonstrate the complexity and the cost of embedding sensors inside a blade and retrieving the data via cables from a





rotary machine. The present work therefore focuses on less intrusive, easier-to-install systems that can provide added value to the wind energy industry at a reasonable price.

## 1.2 Recent developments in microelectronics

Due to the complexity and costs related to embedding conventional aerodynamic and acoustic measurement technology into rotor blades, this present paper focuses on the application of a cost-effective MEMS-based aerodynamic and acoustic measurement system for rotor blades that is thin, non-intrusive, easy to install, low power, self-sustaining and wirelessly transmitting.

In general, recent developments in electronics, wireless communication and MEMS[1] sensors are making it possible to acquire data in a cost-effective and energy-efficient way. Novel IoT [2] sensors are enabling some new and important research areas for many applications, including Structural Health Monitoring (SHM) and predictive maintenance (Di Nuzzo et al., 2021; Chen et al., 2021). SHM aims to detect anomalies and prevent apparatus faults (Chen et al., 2021; Qu et al., 2019) at low cost and connected to a long term vision of improving performance and/or reducing costs of a particular asset. Previous analyses have demonstrated the potential of using inexpensive and low power MEMS sensors aerodynamic purposes (Fathima et al., 2021; Di Nuzzo et al., 2021). For example, arrays of MEMS barometers have already been deployed in other application scenarios, namely to aeroplane wings (Raab and Rohde-Brandenburger, 2020) and to cars (Filipskỳ et al., 2017). In general, cited works show that MEMS sensors are a valid option to acquire aerodynamic measurements. However, none of them address the wireless communication and power consumption challenges required for the continuous monitoring of wind turbines using an IoT device (Karad and Thakur, 2021).

In previous publications, a few examples of wireless devices have been proposed in the wind turbine context (Wondra et al., 2019; Di Nuzzo et al., 2021; Lu et al., 2019). However, they mostly support vibration measurements (Di Nuzzo et al., 2021; Esu et al., 2016) for SHM modal analysis, where the electronics have to process and transmit a data stream in the range of 5 kbps. However, for aerodynamic and acoustic measurements on wind turbine blades, a minimum throughput larger than 1 Mbps is required (Fischer et al., 2021). The data collected by arrays of barometers and microphones is crucial for understanding the aerodynamic and aeroacoustic behaviour, but on the other hand, it poses major challenges in the design of an energy-efficient and long-lasting IoT sensor node. These challenges will be addressed in the present work.

## 1.3 Providing added value to research and industry

As well as helping to further the understanding of three-dimensional, turbulent flow over rotor blades operating in the field under real conditions, pressure and acoustic rotor blade measurements can provide added value to the wind energy community in the following ways:

1. Using the measurements to **infer local angle of attack and rotor inflow conditions** can help wind turbine manufacturers relate rotor blade performance to inflow conditions and enable them to improve their design tools. In previous work, the

---

[1]Micro-Electromechanical Systems

[2]Internet of Things





local inflow conditions have been measured using probes positioned at the leading-edge of the blades and a reference pressure measured in the hub or far upstream, using long tubes (Medina et al., 2011; Troldborg et al.; Wu et al., 2019). However, a reference pressure is not simple to define and to acquire as the free-steam velocity decreases when approaching the wind turbine. Moreover, measurements from pitot tubes must still be corrected to estimate the local inflow conditions. This method is very time-consuming and expensive to apply. Therefore the present work avoids measuring a reference pressure and will not rely on pitot tubes. Instead, the local inflow conditions will be inferred from the pressure gradient at the leading edge (see Section 4.1).

2. The improved understanding gained from the measurements can help OEMs to **improve their aerodynamic and acoustic design tools**, reducing investment costs of wind energy.

3. The improved design tools can then lead to **more efficient and less noisy rotor blade designs**, reducing investment costs of wind energy.

4. The measurements can enable **early detection and classification of local blade surface damage or deterioration**, which can reduce operating costs and increase wind project revenues by improving operators' decision-making regarding blade cleaning and repair. One of the key topics related to this point is Leading Edge Erosion (LEE), which can result from abrasive airborne particles or weather conditions, and can impact the Annual Energy Production (AEP) of a MW-scale wind turbine on the order of 5% (Langel et al., 2015). Current methods for identifying LEE involve manual (Nielsen et al., 2020) or drone-based visual inspection (Shihavuddin et al., 2019), electrical signal analysis (He et al., 2020) or vibration monitoring (Skrimpas et al., 2016), methods which either require the turbine to be shut down or are limited for continuous monitoring (Du et al., 2020). Therefore in the present work, a data-driven model is used to predict the state of degradation of the leading edge of a two-dimensional airfoil via aerodynamic pressure coefficient learning, under the influence of various uncertain inputs and parameters (see Section 4.2).

5. The measurements can enable **increased revenues by improving operators' decision-making and asset management of sub-optimal control settings, blade mass or aerodynamic imbalance**. Existing methods investigated in the literature include detecting imbalances on wind turbine rotors using a harmonic analysis of the rotor response in the fixed frame (Cacciola et al., 2016), using a combination of blade and nacelle measurements, most of which can be obtained from standard instrumentation already found on utility-scale wind turbines (Kusnick et al., 2015), and a combined optimisation of the power and loads using wake redirection by assessing the influence of load variations of the rotor due to partial wake overlap (van Dijk et al., 2016). These methods are all theoretical and have not been proven in the field in a robust manner.

6. The acoustic measurements can enable the **detection of amplitude modulation**, an acoustic effect known to cause annoyance and reduce wind energy project acceptance, which can increase the wind farm operating envelope and thus increase wind project revenues, e.g. Tian and Cotté (2016); Oerlemans and Schepers (2009); Larsson and Öhlund (2014).



7. Measurements on single wind turbines with **retrofit devices** installed (e.g. vortex generators and trailing edge serrations) can allow operators to quantify their effect on performance and thus decide whether to invest in their application to other wind turbines at a site, e.g. De Tavernier et al. (2021); Zhu et al. (2022).

8. The measurements may even allow **early detection of local blade structural damage**, which can reduce operating costs by enabling early repair or decisions to be made (note that this application is still under investigation). For the detection of damage in wind turbines structures, classical vibration- or strain-based monitoring methods rooted in the derivation and the tracking of modal properties have been the prime focus of research (Weijtjens et al., 2016). Clustering approaches are further applied on the Operational Modal Analysis results to reduce the effect of environmental and operational conditions (Oliveira et al., 2018). The identification results could be improved via a modified stochastic subspace system identification, for instance, as proposed by(Dong et al., 2018), or via direct measurements on the blades (instead of tower and nacelle only) (Tcherniak and Larsen, 2013), especially to improve the observability of aerodynamically damped modes for damage detection.

## 1.4 Goal of this work

The goal of this work is to design, test and demonstrate the added value of a prototype cost-effective MEMS-based aerodynamic and acoustic measurement system for rotor blades that is thin, non-intrusive, easy to install, low power, self-sustaining and wirelessly transmitting. This is part of the Aerosense project, which has the ultimate goal of developing pilot measurement systems proven on MW-scale wind turbines.

In this paper, the design of the measurement system is discussed in Section 2, firstly related to the overall system and then to the key sub-systems. In Section 3, the system test and demonstration is described. Finally, two applications of the measurement system that provide added value to the wind energy industry are demonstrated in Section 4: (1) inferring local angle of attack, and (2) detecting and classifying Leading Edge Erosion (LEE). The conclusions can be found in Section 5.





## 2 Design of the measurement system

In this section, the design considerations of the measurement system are first introduced, followed by a description of the design of the overall system and the key sub-systems.

### 2.1 Design requirements

In order to establish the design requirements, the following three priority use cases were first defined based on the expected added value introduced in Section 1 alongside the results of personal interviews with potential customers:

1. **Use case 0: Operational measurement system**: This use case represents a fully functioning measurement chain including collecting the measurement data, pushing it to a digital twin in the cloud, calibrating, correcting, filtering and storing it and finally checking its plausibility. The use case provides value to customers by providing them with the raw data for
evaluating and analysing the behaviour of the operating wind turbine.

2. **Use case 1: Improved aero-acoustic models**: This use case involves expanding use case 0 to include evaluation modules in the digital twin that directly allow the user to compare the results to two-dimensional measurements or simulations and understand the three-dimensional flow effects in field operation, thus improving their aerodynamic and acoustic designs and their design tools. This includes Machine Learning (ML) modules that infer the angle of attack and classify the data
according to external and operating conditions.

3. **Use case 2: Surface damage detection**: This use case involves expanding use case 1 with further ML modules that allow detection and classification of surface damage, in order to help operators optimise performance and make decisions related to maintenance planning.

Based on these use cases, the main requirements used for the initial design included, but were not limited to, the following
considerations:

– The system should be easy to install and remove without damaging the blade, not affect the airflow too much (maximum height $\leq 0.3\%$ of chord, $\Delta C_P \leq 0.3\%$, where $C_P$ refers to the pressure coefficient), and be protected from the weather.

– There should be enough pressure sensors around blade at one radial location to allow the pressure distribution to be obtained: approximately 40 sensors distributed around the pressure and suction sides with a higher resolution near the
leading edge.

– There should be enough acoustic sensors at the trailing edge to be able to estimate parameters used in low-order aero-acoustic models, such as the chord-wise and span-wise correlation lengths as well as the convection velocity and the pressure fluctuation spectrum, with the constraints of being able to transfer the data in the limited bandwidth provided by IoT: the best trade-off has been found with 10 sensors in an L-shape with varying distances between each microphone.




– An Inertial Measurement Unit (IMU) should be present in order to establish the blade position, angle, speed and accel-
         eration.

       – The sampling frequencies of the sensors should be high enough to capture the key dynamics in the system. A summary
         of relevant dynamic effects on a wind turbine and the sampling frequencies of the different sensors chosen for the
         Aerosense system is presented in Fig. 1. Due to the Shannon-Nyquist sampling theorem, the sampling frequency of the
sensors should be at least two times higher than the highest frequency which has to be acquired.

       – The system price should be on the order of $5,000 for the measurement system and $ 50'000/year for a measurement
         service.

       – The lifetime should be on the order of four years, to allow multiple measurement campaigns of several months with one
         system.

– Long-term drift and malfunctioning sensors need to be accounted for.

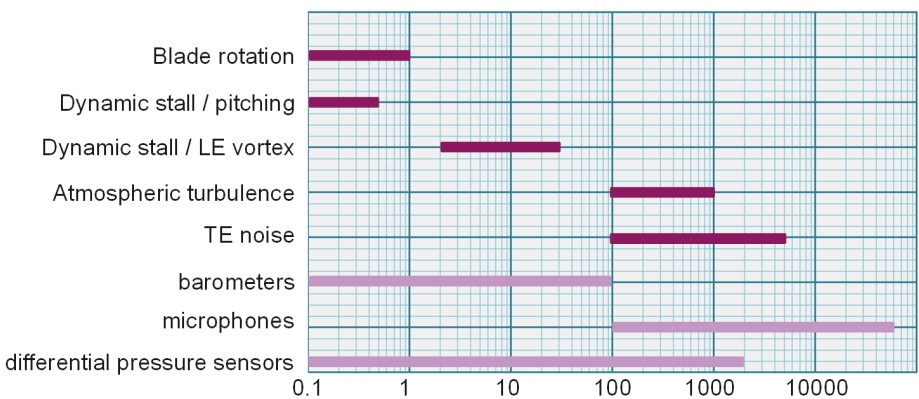

**Figure 1.** Dynamics of the physical features that should be measured by the Aerosense system and the range of frequencies that the sensors
can acquire. Horizontal axis in Hz and in log scale).

## 2.2    Overall system design

An overview of the system is shown in Fig. 2. The system consists of three sub-systems: (1) The sensor node, (2) The base
station and (3) The digital twin on the cloud. In this paper, the design of the sensor node and the digital twin will be described
below. The base station design is ongoing and is not required for the functional tests shown in this paper.

## 2.3    Hardware design of the sensor node

The sensor node consists of a thin sleeve wrapped around the entire blade with embedded MEMS sensors (pressure, acoustic,
inertial, temperature), electronics, power supply and data transmission system. Following the definition of the initial require-



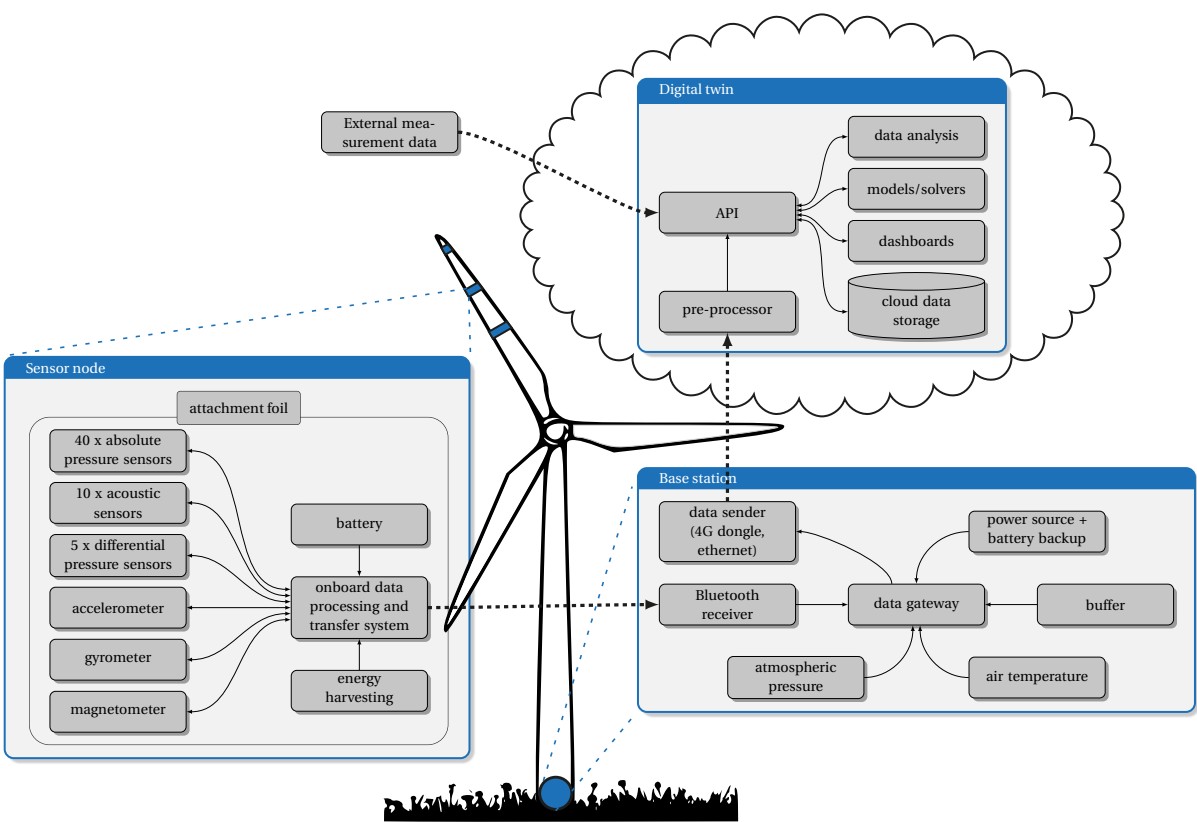

**Figure 2.** Overview of the Aerosense system.

ments described above, an ultra-low power wireless sensor node was designed, tested and verified. In particular, to measure the pressure distribution and the acoustic behaviour on a wind turbine blade, a large number of sensors is needed, which generate

a large amount of data (typically MBs).

In electronics design, this need is in contrast with standard low-power requirements. Hence specific design considerations were made in order to tackle the hardware and software design challenges. Other than a multi-core System on Chip (SoC), the sensor node includes a 512 MB non volatile memory, two external analog-to-digital converter to support up to 10 channels in parallel, and a smart power management system. It includes a cluster-parted power domain distribution and a solar energy

harvester. Moreover, to exploit as much as possible the 1 Mbps bandwidth of the BLE 5.0, an on board data-compression algorithms decreases the amount of data forwarded to the cloud. The system is designed to be able to operate with three nodes connected to one base station. The sensor node supports the following features:

- Long range and low power Bluetooth communication at 1 Mbps and with a maximum coverage above 200 meters.

- Support for up to 40 MEMS absolute pressure sensors (barometers) sampled at 100 Hz, mod. LPS27HHWTR.

- Support for up to 10 wide-range MEMS microphones sampled at 16 kHz, mod. VM 2020.



- Support for up to 5 differential pressure sensors sampled at 100 Hz, mod. Pewatron 52-Series.

- 512 MB on board Flash memory, mod. TC58CYG2S0HRAIJ.

- A MEMS Inertial measurement unit (IMU) sampled at 1 kHz, mod. BMX160 from Bosch.

- On board lossless and lossy compression algorithms.

- Solar energy harvester.

All these properties have been implemented according to the use cases requirements presented above. The final block diagram of the sensor node is shown in Fig. 3.

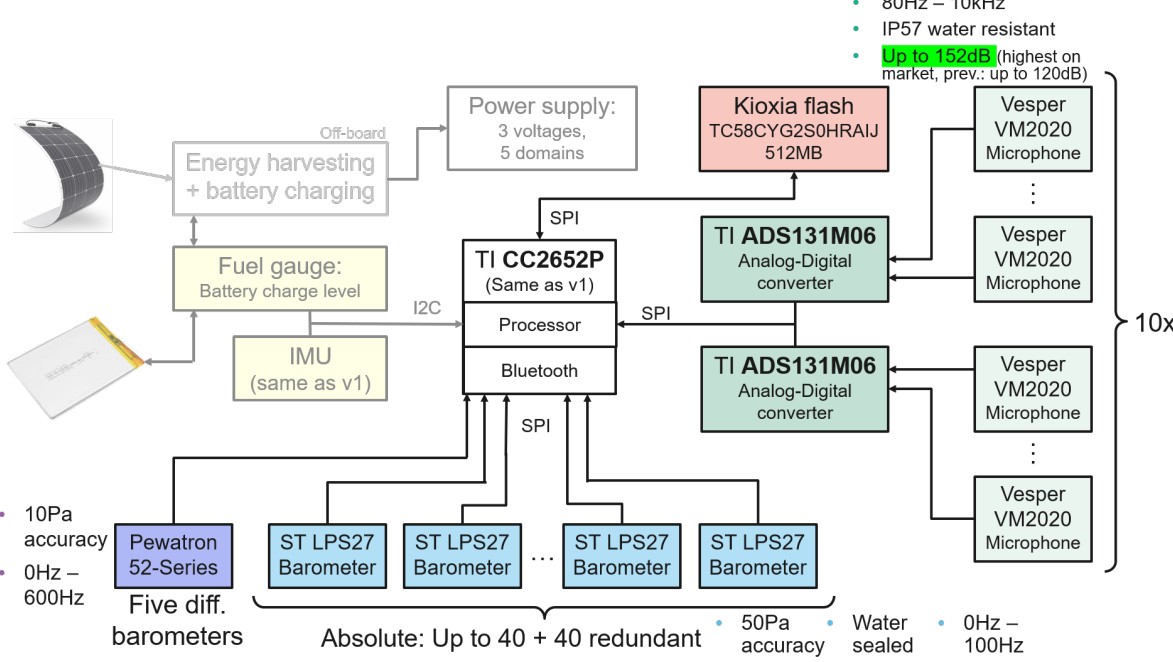

**Figure 3.** Aerosense low power sensor node block diagram

As well as the electronics, the integration of the system in a housing, or sleeve, had to be considered. This sleeve has to enable the system to be easily installed on and removed from a wind turbine blade, protect the electronics from the weather conditions and minimise the aerodynamic impact of the sensors on the flow over the blade. The solution chosen is a custom-made polyjet 3D printing housing, which is flexible enough to bend around any airfoil as shown in Fig. 4, in which the housings were tested for robustness on a 6 kW operating wind turbine. The housing is fixed onto the blade with the same type of adhesion tape that is also used for leading edge protection of wind turbine blades. It is then easy to install by a technician even on mounted blades, it sticks well, and can be removed without damaging the blade.






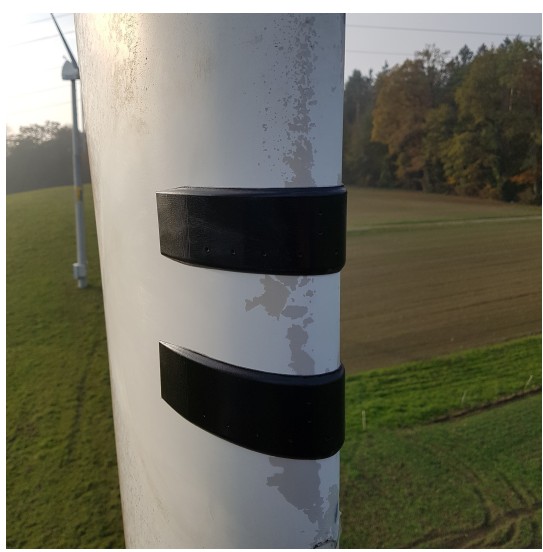

**Figure 4.** 3D polyjet printing housings glued on an operating 6 kW operating wind turbine to test its robustness

## 2.4 Digital twin design

The digital twin system is an essential part of this project, required in order to use the measurement data to provide added value to the customers. As shown in Fig. 2, the software layer of the system includes (a) a **data pre-processor** to collect, timestamp, clean, correct, calibrate and store (in a BigQuery database) the measurement data as well as the external data collected (such as SCADA and met mast data), (b) **inverse problem solvers** to infer quantities such as the angle of attack and the leading

edge erosion class (using e.g. trained ML models) and forward problem solvers to predict non-measured quantities such as the structural deformation of the blade (e.g. using Fluid-Structure-Interaction simulations), (c) **data analysis algorithms** such as a post-processor that computes derived quantities such as the lift and drag coefficients, and (d) **dashboards** to display and download the results.

In order to implement this, the digital twin architecture had to be defined and developed. The software development pipeline

has been set up together with the UK company Octue[3] according to the best industry practice, with git branching/version tracking, testing, documentation and continuous integration / continuous deployment hosted on GitLab. Two examples of inverse problem solvers developed are discussed in Section 4. The existing software packages (forward solvers) inside the digital twin also had to be wrapped for cloud deployment. The wrapping process requires definition of input/output variables and files, as well as configuration variables/files for each software package via json schemas. The wrapped package is then

deployed on the cloud as a service. For the use cases discussed in Section 4, the following software was wrapped:

---

[3]https://www.octue.com/





- OpenFOAM[4]: The software was wrapped via pyFoam[5] and Octue SDK[6] to create a pipeline capable of automatically running 2D airfoil simulations with varying inputs. The inputs are sampled from their presumed probability distributions. The simulated data is used as a train data set for machine learning algorithms and for the purposes of uncertainty quantification.

- Construct 2D Meshing utility: This software creates structured, high-quality 2D airfoil meshes. The modified version of the software developed by Fraunhofer IWES[7] was wrapped with Octue SDK and implemented as a child-process for the OpenFOAM service.

- XFOIL: Python wrapped version of XFOIL called xfoil-python[8], developed by DAR Corporation was further wrapped with Octue SDK and deployed on the cloud. During verification and validation, several bugs have been discovered and fixes were implemented in the xfoil-python GitHub main branch.

- OpenFAST[9]: The software was wrapped using NREL developed python-tools. The basic Octue SDK wrapper was defined to take site wind conditions, generate an inflow data via TurbSim, and perform an aeroelastic simulation with AVENTA WT model.

## 3    Test and demonstration of the measurement system

Following the initial design of the system, the sensor node was built and tested for the first time on a rotating wind turbine model in the small-scale wind tunnel at OST in Rapperswil, Switzerland. The primary goals of these tests were to evaluate how hard it was to design and build flexible PCBs and solder sensors, to evaluate the firmware and the Bluetooth communication in a windy and rotating environment, to extract the first results and see what we could expect, as well as to gain experience with the system.

For these first tests, no measurement sleeve was used as the focus was on the electronics and the communication. A set of sensors (barometers, microphones, IMU) were preliminary chosen, and a microcontroller with a Bluetooth connection and its firmware was developed. The system consists of 40 ST LPS27 absolute pressure sensors and 10 InvenSense ICS-43434 acoustic sensors installed on a flexible PCB. The system was then installed on the blade of a small-scale vertical axis wind turbine inside the OST wind tunnel, with a blade chord length of $70\,\mathrm{mm}$, a blade height of $0.5\,\mathrm{m}$ and rotor radius of $0.35\,\mathrm{m}$ m, as shown in Fig. 5. Measurements were made at a range of wind speed of $0\,\mathrm{m\,s^{-1}}$ to $7.5\,\mathrm{m\,s^{-1}}$ and a rotation speed range of $0\,\mathrm{rpm}$ to $345\,\mathrm{rpm}$, to reach a range of tip speed ratios between 0 and 6.3 and a Reynolds' number based on the incoming wind velocity and on the chord of the airfoil in the order of $10\,000$.

---

[4] https://openfoam.org/

[5] http://openfoamwiki.net/index.php/Contrib_PyFoam

[6] https://github.com/octue/octue-sdk-python

[7] https://gitlab.cc-asp.fraunhofer.de/iwes-cfsd-public/wtrb-aerodynamics/c2d-ext

[8] https://github.com/DARcorporation/xfoil-python

[9] https://www.nrel.gov/wind/nwtc/openfast.html



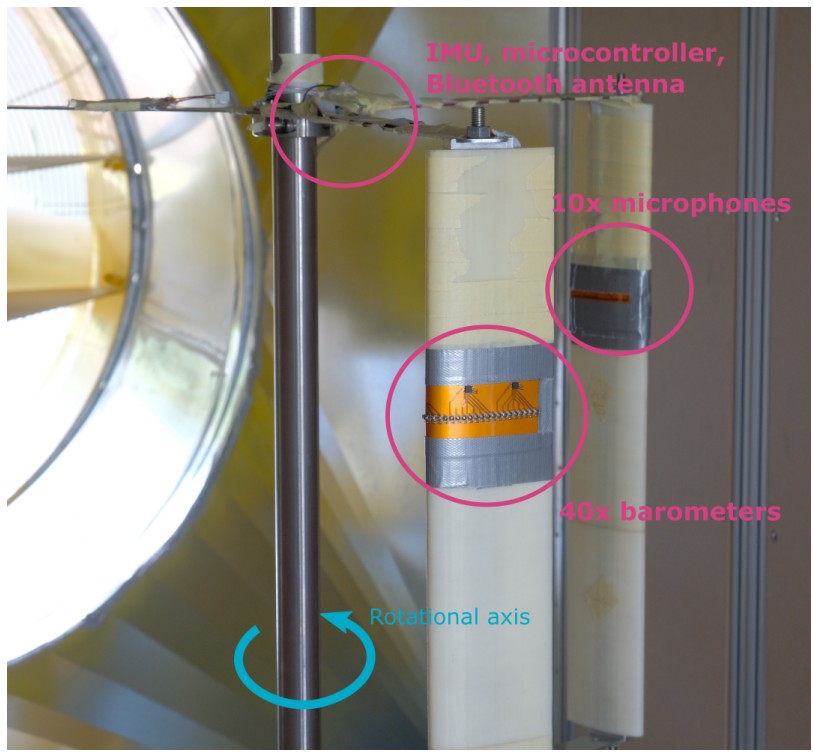

**Figure 5.** Layout of the first version of the Aerosense system on a vertical axis wind turbine in the OST wind tunnel.

A phase-averaged pressure distribution during the complete rotation of one blade is presented in Fig. 6 for a tip speed ratio = 2. In the vertical axis, the zero line represents the leading edge of the airfoil. Negative values towards $-1$ shows the pressure on the inner side of the airfoil, and towards 1, the pressure on the outer side of the airfoil. The horizontal axis represents the position of the blade during the rotation around the vertical axis. The blade starts with the leading edge of the blade facing upwind, and at $\pi/2$, the blade is at its upstream position, while at $3\pi/2$, the blade is in the downwind situation. The position of the blade is known thanks to the IMU of the measurement system. The red colours indicates when and where most the aerodynamic force is generated. As known in the literature (Li et al., 2013; Rossander et al., 2015; Delafin et al., 2017; Barber and Nordborg, 2018) among others, due to a large variation of the local angle of attack of the blade, the suction is mostly on the outer side of the airfoil but is present on the inner side when the blade is in the upstream part. It can be seen that these variations are well captured by the measurement system.



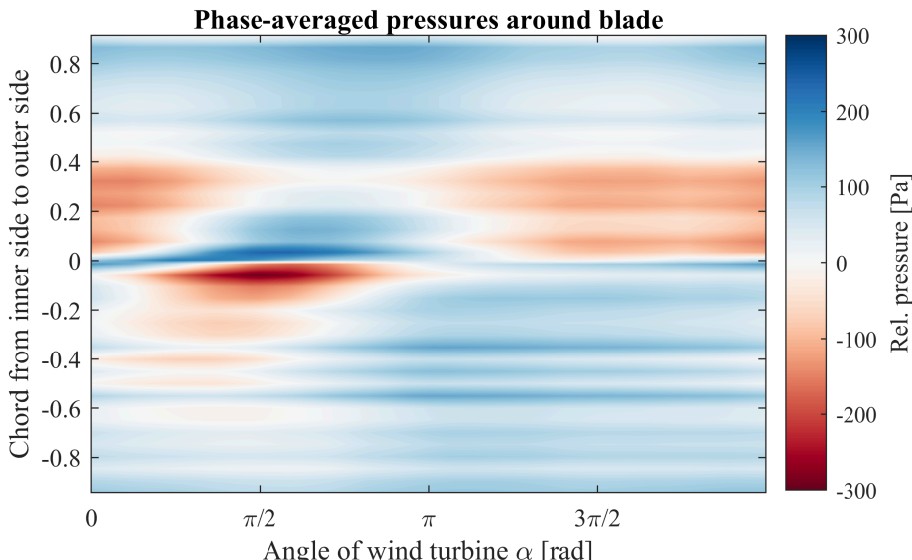

**Figure 6.** Phase-averaged pressures around the blade for a complete turn of the blade of the vertical axis wind turbine at a tip speed ratio of 2. Negative chord is the inside of the blade, zero the leading edge, and positive chord is the outside of the blade.

These tests aimed to evaluate if it was technically possible to capture the main flow features with a wireless measurement system comprising various types of sensors. As the measurement system has not been designed to measure such low pressure variations and for such a small blade without smooth housings (here $70\,\mathrm{mm}$, while blades where the system will be installed have a size in the order of 1 m or more), the physical meaning of the measurements do not have to be investigated in more detail.

The main outcomes of these tests were:

- The measurement system can be installed on an airfoil and can record and transmit data even if it is in rotation with an incoming wind.

- Retrieving the "zero" value (when there is no wind) from the measurement system is of primary importance to achieve accurate results.

- The centrifugal acceleration should be taken into account. Preliminary tests showed there could be a bias of 5Pa/g. On multi-megawatt wind turbines, acceleration can reach $100\,\mathrm{m\,s^{-2}}$, which would affect the sensor measurement by $50\,\mathrm{Pa}$.

The measurement results show that the system is capable of delivering relevant results continuously, although work needs to be done on calibrating and correcting the pressure signals, as well as on refining the concept for the attachment sleeve for weather protection in the field. On-going work involves field tests on an operating 6 kW wind turbine in Switzerland.





# 4 Applications for added value

In this section, two applications of the measurements that could provide added value to the wind energy industry are demon-
strated: (1) inferring local angle of attack, and (2) detecting and classifying Leading Edge Erosion (LEE).

## 4.1 Inferring angle of attack

In this part, a method for deducing the local angle of attack using the measured pressure gradient near the leading edge is
investigated and demonstrated using a set of wind tunnel measurements. The method involves utilising differential pressure
sensors. Small variations of angle of attack or/and wind speed would change the pressure gradient at the leading edge, which can
be captured by differential pressure sensors. Differential pressure sensors do not require the use of a known reference pressure,
commonly located far upstream or in an area with no wind. While it can easily be done in controlled environment, such as in a
wind tunnel, a known reference pressure is much more complicated to acquire on a wind turbine. Differential pressure sensors
work on a smaller measuring range than absolute pressure sensors and are therefore more sensitive are are able to detect smaller
variations of pressures and therefore angles of attack. In the Aerosense system, the differential pressure sensors use the same
reference pressure point, $P_0$, which is an arbitrary point located in the leading edge region (Fig. 8). Discrete values of pressure
difference at the leading edge with an arbitrary point $(P_i - P_0)$ are used to interpolate the pressure gradient due to a specific
incoming flow (angle of attack and wind speed) passing around the leading edge (Fig. 9). The measured variations of pressure
are fed into an algorithm based on a potential flow model passing a parabola (Saini and Gopalarathnam, 2018) to estimate the
angle of attack and the incoming flow velocity without the need of external measurements and reference pressure.

The feasibility of the method has been demonstrated on a 3D-printed NACA0018 profile section in the sub-sonic wind tunnel
at ETH Zurich, Switzerland, as shown in Fig. 7. The airfoil was designed and 3D-printed specifically for these tests with a chord
of $25\,\mathrm{cm}$ and a span of $1\,\mathrm{m}$. A set of 40 flush pressure taps were integrated at mid-chord for reference, with corresponding
digital pressure transducers. The blade was designed to be tilted $10°$ backwards to mimic a span-wise component of the flow
similar to what could be found on a wind turbine blade, and was fitted with three different barometer strips installed and sand
paper at the leading edge. The measurements were used for various other tests and comparisons within the Aerosense project.
Loads exerted on the blade were acquired by a 6 degrees-of-freedom balance. In order to test the method of inferring the angle
of attack, tests were made within a static angle of attack range of $-30°$ to $30°$, at three different wind speeds (10, 30 and
$50\,\mathrm{m\,s^{-1}}$, resulting in a range of Reynold's number based on the chord of the airfoils between $2 \times 10^5$ and $8 \times 10^5$.

Figure 9 presents the pressure values in the leading edge region in the first 10% of the chord, from the 40 pressure taps
with an arbitrary $P_0$ reference point. The horizontal axis $\eta$ corresponds to a non-linear curvilinear axis linked to the parabola
fitting (Ramesh, 2020), which enhances the gradient of pressure at the leading edge. The black thick line is created thanks
to the values of five chosen pressure points, shown in colour in Fig. 8 and Fig. 9, that are used to fit the analytical pressure
distribution of a flow passing a parabola. The measured pressure distribution at the leading edge can be well represented by an
inviscid flow passing a parabola.





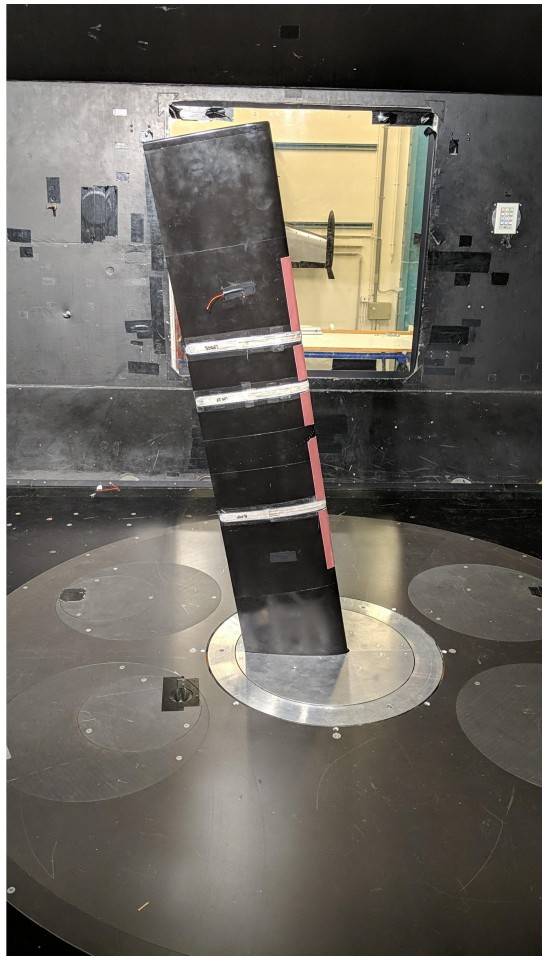

**Figure 7.** 3D printed NACA0018 blade installed in the test section of the ETH Zurich wind tunnel. The blade is in the 10° tilted configuration, with three barometer strips installed and sand paper at the leading edge.

This pressure distribution passing through a parabola depends on the stagnation point and the flow velocity. From the stagnation point, it is then possible to retrieve the angle of attack in a look-up table.

     This stagnation point method has the advantage of being applicable to three-dimensional flow typical on operating wind turbines. Traditional methods of obtaining or inferring the angle of attack such as inverse BEM method, or 3-point method (Vimalakanthan et al., 2018), are limited by the fact that the definition of angle of attack involves assuming that the flow

remains inside a two-dimensional plane, as is the case for wind tunnel tests. On a rotating wind turbine, however, the flow is mostly three-dimensional, hence a true angle of attack in this sense cannot be defined. Moreover, the incoming flow gradually decreases approaching the blade due to the induction, hence it is not possible to assess a "true" incoming wind velocity as it is the case in a wind tunnel, and to build a correct vectorial combination to evaluate the relative wind speed on the rotor blade.





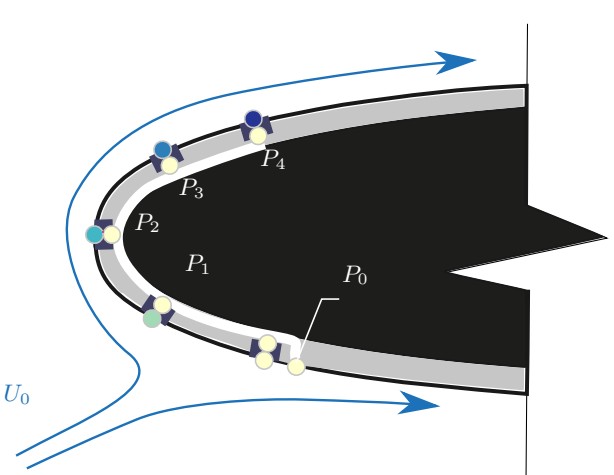

**Figure 8.** Sketch of differential pressure sensors at leading edge. The reference pressure $P_0$ is taken at a point in the leading edge area on the pressure side. Four other sensors record the difference of pressure between their locations and the reference pressure.

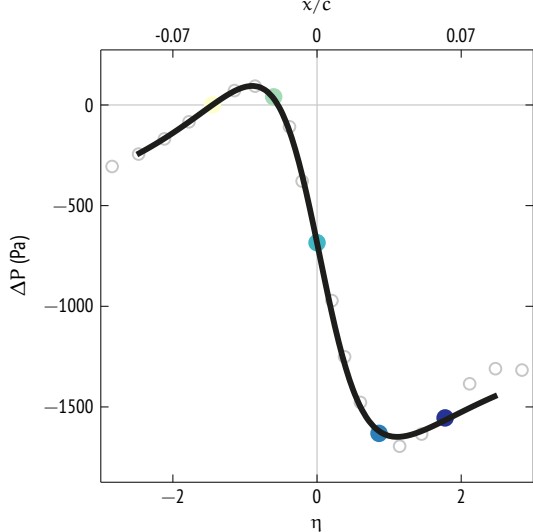

**Figure 9.** Pressure distribution at the leading edge. Coloured points represent the values from the differential pressure sensors shown in Fig. 8. The other empty circles represent values from other pressure points. The thick black line is the result of an inviscid flow passing a parabola modelling the leading edge.

The results are shown in Fig. 10, in which the position of the stagnation point based on the inviscid flow model using only five differential pressure points in the first 5% of the chord for different wind speeds can be seen. For comparison, the location of maximum of pressure, which is the dynamic pressure at the stagnation point, has been found using the 40 flush pressure taps at the mid-chord of the airfoil (see crosses in Fig. 10). Our method finds the same stagnation point position as the pressure taps, even when the flow starts to detach near the trailing edge (above 10°). The method is not as accurate for angle of attacks larger than the stall angle of the airfoil, as expected. For wind turbine applications, the optimal working range of angle of attack is below the stall angle, where this method performs well.

From the stagnation point, it is possible to retrieve the angle of attack using a look-up table built with XFOIL in this case (Fig. 11). The estimated angle of attack is slightly lower than the actual angle of attack set in the wind tunnel, if the stagnation point positions found in XFOIL and with pressure taps are compared. More precise measurements and corrections on the experimental data as well as more accurate simulations would probably help reducing this gap. However, as long as the angle of attack is not greater than the stall angle, the error is not larger than $\pm 2.5°$, and would be satisfactory for a non-intrusive method based on MEMS differential pressure sensors.

Further experiments were carried out with a roughened leading edge and a tilted wing, in order to test the robustness with leading edge erosion and with three-dimensional flow. The estimation of the stagnation point position remained as precise as





with a clean airfoil. This angle of attack estimation method seems therefore adequate for wind turbine purposes. On-going

work involves testing the system in the field.

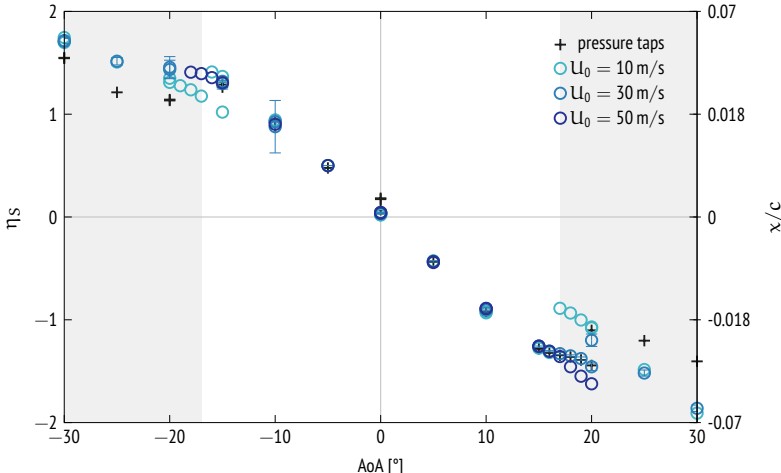

**Figure 10.** Position of the stagnation point for different angle of attacks at different wind speeds on a NACA0018 based on 5 differential pressure measurements at the leading edge (circles) and by finding the maximum of pressure using 40 flushed pressure taps at $30\,\mathrm{m\,s^{-1}}$ (crosses). Shaded areas show the stall region of the airfoil.

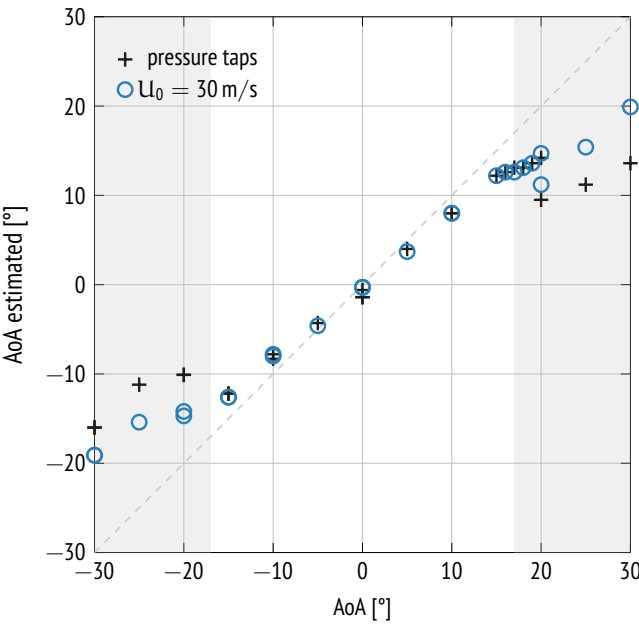

**Figure 11.** Estimation of the angle of attack using our method and a look-up table. Comparison with estimation of the angle of attack from pressure taps are shown with the crosses.

## 4.2 Detecting and classifying Leading Edge Erosion (LEE)

In this part, data-driven methods for learning to diagnose LEE on an airfoil via aerodynamic pressure coefficients are investigated and demonstrated using aeroelastic simulations and Computational Fluid Dynamics (CFD). The existing measurements could not be used for the demonstration because measurements are not available for different levels of LEE from this measure-
ment campaign.

Two approaches can be taken to diagnose the severity of LEE using the aerodynamic output streaming from the Aerosense device. A first approach involves the use of time-series of integrated pressure quantities (lift and drag), while the second utilises instantaneous snapshots of the pressure field. In essence, the first approach focuses on the temporal component, while the second concentrates on the spatial aspect.

For the first approach, a necessary prerequisite is the modelling and simulation of the aeroelastic response of a wind turbine undergoing blade LEE. In order to do this, a pipeline was developed as follows. The time-dependent degradation process affecting the aerodynamic properties of a blade is modelled with a Non-Homogeneous Compound Poisson Process, a stochastic process which parametrises the cumulative damage on a blade section caused by the arrival of random degradation-inducing shocks. The aeroelastic response to this degradation is then simulated by coupling this process to OpenFAST (an aeroelastic
wind turbine simulator), under uncertainty in the environmental conditions. This simulation pipeline is used to generate a dataset of aeroelastic time-series data (wind inflow velocity, angle of attack, lift and drag coefficients) corresponding to different



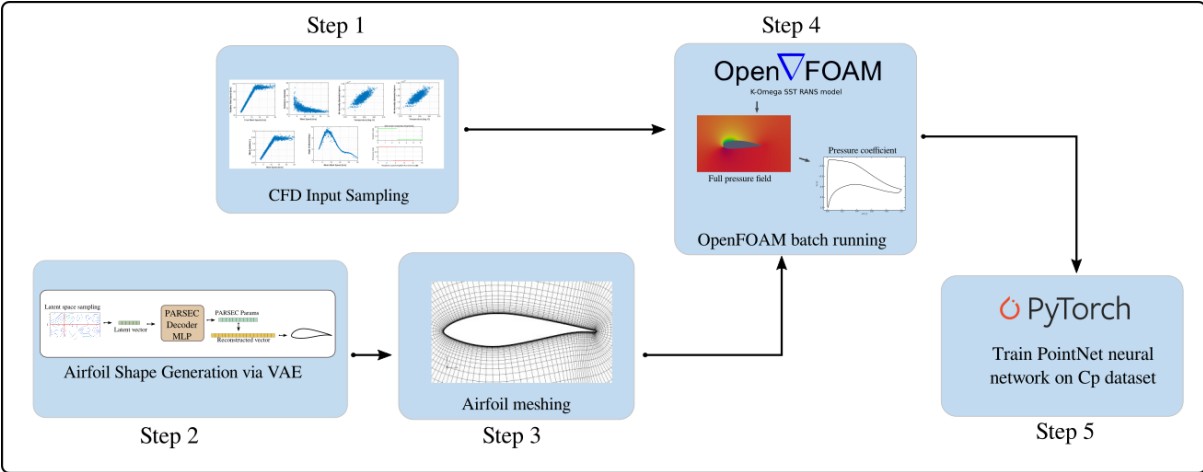

**Figure 12.** Overview of the proposed approach to simulation aerodynamic pressure coefficient data under diverse conditions of LEE, which is then used to train a neural network for diagnostics purposes.

categories of LEE severity. Then, a Transformer neural network (Vaswani et al., 2017) is trained on this database in a supervised manner, such that when it is given a multivariate time-series input, it outputs a prediction for the severity of LEE. More information on this approach can be found in Duthé et al. (2021).

The second approach also involves computationally generating training data, albeit via CFD simulations in this case, as shown in Fig. 12. In order to create a robust training dataset which encompasses a reasonably wide variety of operational conditions, in the first step the 2D CFD input parameters (inflow velocity, turbulence intensity, angle of attack, roughness height, chordwise extension of LEE roughness) are modelled as probabilistic variables. A suitable distribution is formulated for each variable, for instance the inflow velocity follows a Weibull distribution and further accounts for wind turbine RPM,

while the distribution for roughness parameters are devised based on the work of Sareen et al. (2014). A preliminary dataset is established with around 300 unique combinations. Additionally, a variational auto-encoder (Kingma and Welling, 2013) is trained on the UIUC airfoil database (Selig, 1996) to construct a distribution of airfoil shapes (step 2), which can then be sampled from in order to generate realistic, yet unique, airfoil shapes to be used as the basis for the simulation meshes (step 3). Samples are then drawn from both the distributions for the flow conditions and for the airfoil shapes, such that a database

of unique CFD simulation inputs is created. In step 4, each simulation is then executed in OpenFOAM using a k-Omega SST RANS model with modified rough wall functions, as suggested in (Knopp et al., 2009). Airfoil surface pressure coefficient data is extracted from the converged simulations and stored along with the corresponding roughness label, thus forming the dataset upon which an adapted point cloud neural network (Qi et al., 2017) is trained in a supervised manner in step 5. The resulting algorithm is able to output predictions for leading edge roughness, given an input of surface pressure coefficients.



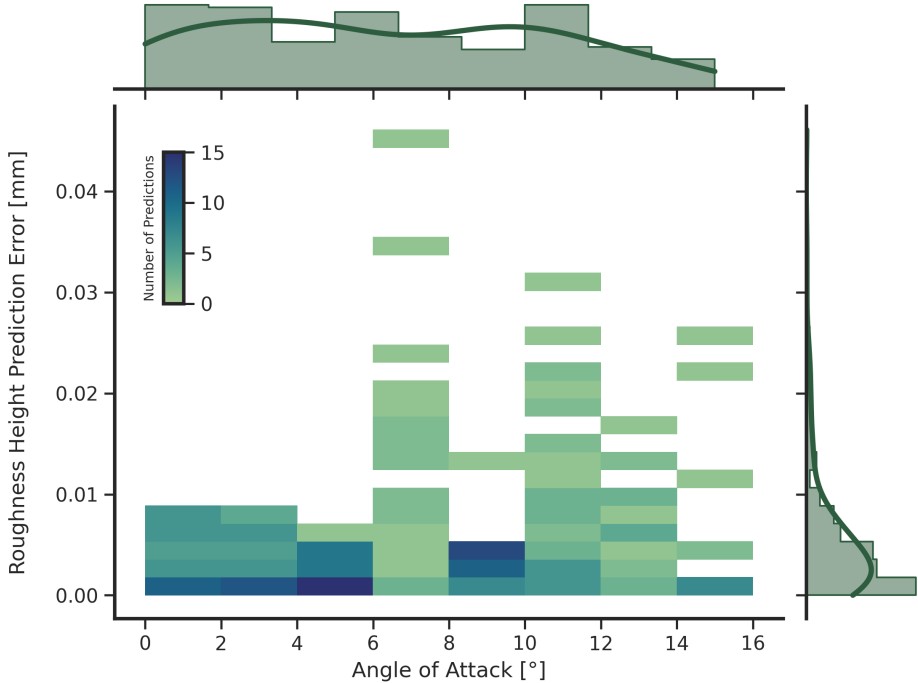

**Figure 13.** Error in the prediction of equivalent sand grain roughness height as a function of angle of attack gathered on the validation dataset. Prediction quality appears to decrease for higher angles of attack.

The results show that through this method, it is possible to estimate the leading edge roughness, given the pressure distribution on the airfoil surface. Preliminary findings indicate that prediction quality is dependent on the angle of attack, as highlighted in Fig. 13. This figure plots the distribution of roughness prediction errors against the angle of attack for the validation dataset. It can be seen that for higher angles of attack, the number of predictions with large errors increases. This aligns well with our expectations: at high angles of attack, minor perturbations in the pressure field caused by increased leading edge

roughness may be secondary to fluctuations induced by a detached flow. Furthermore for situations with small amounts of leading edge erosion, we notice large relative prediction errors. This outcome is also in line with our presumptions, as in these cases we expect that the flow will only be modestly affected by small increases in leading edge roughness.

    Further work in this matter should focus on extending these methods to real experimental data measured by the Aerosense system, as it becomes available. Furthermore, mitigation strategies and different architectures should be explored to overcome

the challenges in predicting erosion for situations with low degradation severities or at high angles of attack. One class of architectures that could be well suited for this use case are Graph Neural Networks (Scarselli et al., 2008; Pfaff et al., 2020), as these network structures enable the representation of positional information, but also allow for temporal updating and therefore dynamics.



# 5 Conclusions

A cost-effective MEMS-based aerodynamic and acoustic wireless measurement system that is thin, non-intrusive, easy to install, low power, and self-sustaining has been designed based on a set of requirements and use cases obtained from interviews with the wind energy industry. The system consists of three sub-systems: (1) the sensor node, (2) the base station and (3) the digital twin on the cloud.

The sensor node includes a long range and low power Bluetooth communication at 1 Mbps with a maximum coverage above 385 200 meters, support for up to 40 MEMS absolute pressure sensors (barometers) sampled at 100 Hz, up to 10 wide-range MEMS microphones sampled at 16 kHz, and up to 5 differential pressure sensors sampled at 100 Hz, as well as 512 MB on board Flash memory, a MEMS IMU sampled at 1 kHz, on board lossless and lossy compression algorithms as well as a solar energy harvester. The electronics are embedded into a custom-made polyjet 3D printing housing, which is fixed onto the blade with the same type of adhesion tape that is also used for leading edge protection of wind turbine blades.

The software layer of the digital twin system includes a data pre-processor to collect, timestamp, clean, correct, calibrate and store the measurement data as well as the external data collected, inverse problem solvers to infer quantities such as the angle of attack and the leading edge erosion class and forward problem solvers to predict non-measured quantities such as the structural deformation of the blade, data analysis algorithms that compute derived quantities such as the lift and drag coefficients, as well as dashboards to display and download the results.

The sensor node was built and tested for the first time on a rotating wind turbine model in the small-scale wind tunnel at OST in Rapperswil, Switzerland. The results show that the system is capable of delivering relevant results continuously, although work needs to be done on calibrating and correcting the pressure signals, as well as on refining the concept for the attachment sleeve for weather protection in the field.

Finally, two methods for using the measurements to provide added value to the wind energy industry were developed and 400 demonstrated. A method for inferring local angle of attack via stagnation point detection using differential pressure sensors near the leading edge was shown to work well for this application via a measurement campaign on a 2D NACA0018 airfoil in a wind tunnel. A method for detecting and classifying Leading Edge Erosion using instantaneous snapshots of the measured pressure fields was shown to be promising using a set of CFD data.

On-going work involves field tests on an operating 6 kW wind turbine in Switzerland, as well as the expansion of the entire 405 system for additional use cases.

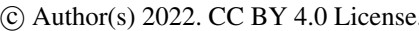

*Code availability.* The code being developed within this project will be made available at a later date

*Data availability.* The wind tunnel measurement data is being analysed and post-processed and will be disseminated in a separate publication.

*Author contributions.* Sarah Barber leads the Aerosense projects and was responsible for writing and coordinating the overall paper. She is
also the co-supervisor of the PhD student Yuriy Marykovskiy. Julien Deparday is a postdoc at OST and was responsible for the wind tunnel
tests and angle of attack method. Yuriy Marykovskiy is a PhD student at OST and ETHZ CSMM and was responsible for the digital twin part
of the project. Eleni Chatzi is the co-supervisor of Yuriy Marykovskiy and supervisor of Gregory Duthé, and was responsible for coordinating
the activities as ETHZ CSMM. Imad Abdallah is a postdoc at ETHZ CSMM and lead the leading edge erosion work. Gregory Duthé is a
PhD student at ETHZ CSMM and carried out the pressure coefficient learning activities for the leading edge erosion detection task. Michele
Magno is head of ETHZ PBL and was responsible for coordinating the hardware development activities. Tommaso Polonelli is a postdoc
at ETHZ PBL and was responsible for designing the sensor node. Raphael Fischer was a Master's student at ETHZ PBL and prepared and
carried out the wind tunnel tests together with Julien Deparday. Hanna Mueller is a PhD student at ETHZ PBL and was responsible for the
data compression activities as well as the hardware design.

*Competing interests.* There are no competing interests

*Acknowledgements.* This work is funded by the BRIDGE Discovery Programme of the Swiss National Science Foundation and Innosuisse,
project number $40B2 - 0\_187087$. Thanks to Justin Lydement and Ueli Spalinger for their support with the field and wind tunnel tests.



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
