# Peer review of "Development of a wireless, non-intrusive, MEMS-based pressure and acoustic measurement system for large-scale operating wind turbine blades"

_Wind Energy Science, 2021_

## Author Comment (AC1)

wes-2021-157

Response to Anonymous Referee #1

Thank you for taking the time to review this paper. We hope that you are happy with our changes, which are marked in red in the mark-up file.

Line 178

It is not always clear, if the verification was in lab or in situ.

We agree that the description in the abstract is not clear enough, and have changed the abstract correspondingly.

Line 244

0.35m m -> 0.35mm

Actually, we mean 0.35m. We have deleted the extra "m".

In general, it is a well written paper, which is worth being published to the scientific community.

As the system will enlarge the thickness of the blade profile, the definition as a non-intrusive system seems to be a bit irritating. Surely it will be possible to determine the influence of the system, but it will have an influence on the aerodynamics, which is not completely negligible.

We understand your irritation with the use of the word "non-intrusive" and agree that the measurement system isn't completely non-intrusive if the meaning of the word is taken 100% literally. We were actually referring to the fact that there is no intrusion into the wind turbine system – there is no electrical connection with the wind turbine and no mechanical intervention is required (the node can be attached and removed without damaging the blade). However, it is true a very small effect on the aerodynamic and acoustic behaviours of the blades is expected, as you mention. We aimed to keep this effect as small as possible by keeping to the requirements given by potential customers, who specified a maximum measurement node thickness of 4 mm.

We have made the following changes:

- We have clarified the intended meaning of the word "non-intrusive" in the abstract by adding a sentence "The measurement system does not require an electrical connection to the wind turbine and can be mounted and removed without damaging the blade."
- We have clarified the topic in the list of requirements on lines 153-155.

Calculating the bandwidth with the named sensors and sample frequencies, it seems that you will need a minimum bandwidth of 2,7 Mbit/s at 16Bit or 5,4 Mbit/s at 32Bit for uncompressed, continuous measurements, which is more than your given max. bandwidth of 1 Mbit/s.  If you found a method to reduce the data, you should point this out.

Thanks for this comment. Indeed, the system description was not sufficiently clear. We updated the text on lines 189-192 specifying how the data is internally managed. However, we believe that in this specific paper any further detail of the compression algorithm goes beyond its scientific contribution.

---

## Author Comment (AC2)

wes-2021-157

Response to Anonymous Referee #2

Thank you for taking the time to review this paper. We hope that you are happy with our changes, which are marked in blue in the mark-up file.

Interesting paper outlining the development of new valuable apparatus to measure aerodynamic pressures on wind turbine blades.

General comments:

It is recommended to pay more attention to careful unambiguous dissemination. I have tried to point out some points in the specific comments but it remains the responsibility of the authors to convey the message across to the general public.

- Thanks for this helpful feedback! Based on your specific comments and on a thorough read of the paper, we modified the paper to avoid any misinterpretation.

Although some wind tunnel tests are described, the aerodynamic effect of adding this system to a specific sections remains unclear. Such a verification should not be to difficult to perform by comparing to conventional taps in the wind tunnel.

- This is true, although not within the scope of this paper. We have added comments to reflect this both in Section 2.1 and at the end of Section 4.1.

Specific comments

Introduction p2

Perhaps it is useful to commemorate historical efforts in IEA Task 14 and 18 that also featured some field test including pressure measurements. From IEA Task 47 it is known that also at DTU there is an ongoing effort to develop a pressure belt, would it make sense to refer to this development as well and identify differences and synergies?

- In Section 1.1, we have added mentions of older measurements as well as a comment about existing efforts that are similar.

-1.2 p3 line 53/55 MEMS footnote on line 55 is introduced after its first usage on line 53

- Thanks for pointing this out – we changed it.

-1.2 p3 line 55 Explain the relevance of IoT in this context

- We have added a comment explaining this.

-1.3 p3/4 The summation of added value items is often overlapping and inter-related. Perhaps restructure or remove enumeration

- Good point – we removed the enumeration.

-2.1 p6 line 152/153 Can something be mentioned about the effect of transition on Cp? It would be worthwhile to mention absolute thickness of the sleeve in mm.

- We have included a comment about capturing the effects such as transition and separation on Cp.
- We have mentioned the absolute thickness here (in red because the other reviewer asked the same thing) as well as on line 179.

-2.1 p6 line 154 Is the mentioned nr of 40 sensors based on a criterium, e.g. accuracy in Cl?

- Yes, we have added this.

-2.1 p6 line 160 Can a reference and/or graphic be given to further illustrate the L-shape configuration mentioned?

- We added a figure and an explanation (Figure 2).

-2.1 p7 Fig. 1. The figure suggests dynamic stall phenomena to occur only below 1 Hz which does not make sense

- We agree that this was unclear, and have added a comment to section 2.1 clarifying this.

-2.1 p7 Fig. 1. Why is the range of dynamic pressure sensors limited to only 2k (perhaps to limit bandwith?)?

- The frequencies of the sensors (barometers, microphones and differential pressure sensors) come from the specifications of the sensors we chose for Aerosense. They were chosen as a compromise between sampling frequency, accuracy, reliability, weatherproofing, and size. 2kHz is the maximum we can obtain with the differential pressure sensors we have.
- We added a comment to section 2.1 to clarify this.

-2.4 p10 line 219 clarify or give reference to json schemas

- Sorry, we added a footnote.

-3 p13 line 258 and beyond, Fig. 6.

It is mentioned that the tests are aimed to evaluate if it was technically possible to capture main flow features with the system. On the other hand it is mentioned the physical meaning of the measurements do not need further investigation. This seems a bit contradictory, perhaps this coudl be rephrased or e.g. the azimuthal load variation from the integrated pressure distributions could be added to verify if the results make sense.

- We rephrased this to make it more clear.

-4.1 p14

Clarify the spanwise position of the belt compared to the 40 taps (or did I miss it?).

- Sorry, the position of the sensors has been added to Section 4.1.

-4.1 p14 line 292. Is mid-span meant instead of mid-chord? (also on p16 line 317)

- Yes, we changed this!

-4.1 p15 line 311. It is claimed that flow on a rotating wind turbine blade is mostly three-dimensional which is a motivator for the subjected approach, is there a reference to substantiate this claim? p16 line 321 then mentions XFOIL is used which is a 2D tool which seems a bit strange in this respect?

- We did not mean to give the impression that "flow on a rotating wind turbine blade is mostly 3D". It is an effect that can occur at certain locations and under certain conditions, and 2D assumptions need correcting for it in order to improve the accuracy of models.
- In order to avoid this false impression, we have reworded a couple of sentences in this paragraph.

-4.1 p16 Fig. 9  Clarify significance of horizontal axis label eta

- We added the link between the chordwise coordinates x with eta to Section 4.1.

-4.1 p16 line 325 An error in angle of attack of 2.5deg can be interpreted as quite large, but is mentioned to be satisfactory. For which application is this the case?

- The 2.5deg inaccuracy might be sufficient to have a general estimate of the angle of attack, but might not be sufficient when finer comparisons are required. The 2.5deg error mostly come from at high angle of attack using a standard look-up table between the stagnation point and the angle of attack near stall made by XFOIL. We strongly believe that a more accurate relationship between the angle of attack and the stagnation point with better simulations or with wind tunnel data would bring more accurate data. For example, in figure 10, the estimation of the stagnation point from the flush mounted pressure taps and the method using the differential pressure at the leading edge has a difference of less than half of a degree.
- We modified the paper accordingly.

-4.1 p17 Fig. 10 Clarify wind speed for pressure taps in legend. Are these results for a tilted or non-tilted blade?

- These results presented here are shown for a non-tilted blade. We added a comment to Section 4.1.

-4.1 p17 Fig. 10. The caption mentions flushed pressure taps, perhaps flush mounted taps are meant here?

- Yes, we changed it.

-4.2 p18 If I understand correctly the sensors are added in a sleeve to be wrapped around the section (This aspect should be clarified better in the text describing the apparatus or did I miss it?). How would such a sleeve impact the erosion measurements as it is wrapped over the eroded surface?

- Yes, you are right, and we explain that right at the beginning of Section 2.3, as well as on lines 208-212. However, perhaps our inconsistency of the words "housing" and

"sleeve" lead to this confusion. We have tried to solve this by only using the word "sleeve".

- The second question is a very good point that we forgot to mention. We have inserted a comment on this on line 350.

-4.2 p18 line 342. Perhaps add a reference for 'Non-Homogeneous Compound Poisson Process'

- Done.

-4.2 p19 line 349. Are any results given of this approach?

- In the paper we refer to, there are many results. We have added a couple of sentences summarizing the results here.

-4.2 p19 line 362. is->are

- The use of "data" as a plural or singular seems to be a highly debated topic at the moment. We chose to use it in the singular in this paper, so will leave "is" if you don't mind. We will ask the editor their opinion on the matter.

-4.2 p19/20 For the second approach it is not clear how this method would work in the field, where sectional inflow conditions are unsteady and unknown due to atmospheric turbulence.

- We have added a few sentences about this (lines 401-405).

-4.2 p20 Fig. 13 The caption should indicate the significance of ?histogram? that is added to the vertical and horizontal axes

- We have modified the caption of Fig. 13 to showcase the significance of the two axis histograms.

---

## Referee Report (RR1)

Reviewer: Alejandro Gomez Gonzalez – Siemens Gamesa Renewable Energy

**Generic comments:**

- Nice work!! Nicely written paper, easy to read and follow, and very interesting and relevant topic
- The applicability of the LEE detection is a bit doubtful from a practical perspective: the pressure belt would probably erode quicker than the blade, and if it is installed post-erosion, then there is not really a point in detecting it. From a scientific perspective it is still interesting in terms of detecting variations in the Cp distribution which might arise from factors other than erosion (e.g. soiling or ice build up)
- The use case of "early detection of local blade structural damage" is not obvious from the current setup
- The "zero" value as mentioned in section 3 will be of high importance, I did not very clearly see how this is done for measurements in the field (I guess that is part of the ongoing work mentioned in line 286?)

**Small comments:**

- Line 111: stating that all these methods are theoretical is maybe a bit too blunt. Most manufacturers will have methods (proven in the field) to detect rotor imbalance.
- Line 116: the quantification of the impact of performance would normally be done via side-by-side analysis, because the differences in power will be so small (se example reference below)

**Relative and Integral Wind Turbine Power Performance Evaluation**

January 2004

Authors:

[Figure]

**Axel Albers**
Deutsche WindGuard Consulting GmbH

- The abbreviation LEE and its full wording "leading edge erosion" is used multiple times across the document (would be enough just once, and then just using the abbreviation)
-

**Typos:**

- Issue with reference Hansen1993 in line 32 of page 2
- Line 49: change "thanks to tubes" → "with tubes" or "via tubes"
- Correct grammar of line 80: "As well as helping to further the understanding of.." → "As well as helping to further understand the.." or "As well as helping the further understanding of.."
- Line 124: space missing in "by(Dong et al"
- Label of Figure 2: there is an opening parenthesis missing

- Line 264 and 314: "Reynolds" instead of "Reynold's" (his last name was Reynolds.. not Reynold 😊 )
- Line 371: check correct use of capital letters in "Transformer neural network"
- The use of capital letters in the title of publications in the reference sections is inconsistent (some of them use capitals, some don't): might be pointed out by the editorial check prior to publication? (not critical)

---

## Author Response (AR2)

**Authors' response to wes-2021-157-referee-report-1**

18.05.2022

**Generic comments:**

• Nice work!! Nicely written paper, easy to read and follow, and very interesting and relevant topic

• The applicability of the LEE detection is a bit doubtful from a practical perspective: the pressure belt would probably erode quicker than the blade, and if it is installed post-erosion, then there is not really a point in detecting it. From a scientific perspective it is still interesting in terms of detecting variations in the Cp distribution which might arise from factors other than erosion (e.g. soiling or ice build up)

Agreed, it isn't feasible to use the pressure belt continuously on the blade. However, because it is easy to install and remove, we could install it for a short period every year or so. This is part of on-going work.

• The use case of "early detection of local blade structural damage" is not obvious from the current setup

We are in the process of publishing some first papers about this to make it more clear. We have a presentation at TORQUE2022 in a few weeks ☺

• The "zero" value as mentioned in section 3 will be of high importance, I did not very clearly see how this is done for measurements in the field (I guess that is part of the ongoing work mentioned in line 286?)

This is true, and we didn't mention how we will do this in the field. We have added a sentence to line 285.

Small comments:

• Line 111: stating that all these methods are theoretical is maybe a bit too blunt. Most manufacturers will have methods (proven in the field) to detect rotor imbalance.

We completely agree and have removed that comment / changed "not proven" to "most have not yet been proven"….

• Line 116: the quantification of the impact of performance would normally be done via side-by-side analysis, because the differences in power will be so small (se example reference below)

True, we have added that measurements may be more accurate than this.

• The abbreviation LEE and its full wording "leading edge erosion" is used multiple times across the document (would be enough just once, and then just using the abbreviation)

Agreed, and changed.

Typos:

• Issue with reference Hansen1993 in line 32 of page 2

• Line 49: change "thanks to tubes" → "with tubes" or "via tubes"

• Correct grammar of line 80: "As well as helping to further the understanding of.." → "As well as helping to further understand the.." or "As well as helping the further understanding of.." • Line 124: space missing in "by(Dong et al"

• Label of Figure 2: there is an opening parenthesis missing

• Line 264 and 314: "Reynolds" instead of "Reynold's" (his last name was Reynolds.. not Reynold

)

• Line 371: check correct use of capital letters in "Transformer neural network"

• The use of capital letters in the title of publications in the reference sections is inconsistent

(some of them use capitals, some don't): might be pointed out by the editorial check prior to

publication? (not critical)

All typos changed, thank you for these observations!